# A Novel MoS$_2$/TiO$_2$/Graphene Nanohybrid for Enhanced Photocatalytic Hydrogen Evolution under Visible Light Irradiation

Tsung-Mo Tien * and Edward L. Chen *

Coastal Water and Environment Center, College of Hydrosphere Science, National Kaohsiung University of Science and Technology, Kaohsiung City 81157, Taiwan
* Correspondence: tmtien@nkust.edu.tw (T.-M.T.); edwardljchen@gmail.com (E.L.C.)

**Abstract:** Graphene is regarded as a potential co-photocatalyst for photocatalytic hydrogen (H$_2$) evolution, but its great photocatalytic ability requires tuning the band gap structure or design morphology of composites. In this study, MoS$_2$/TiO$_2$/graphene (MTG) nanohybrids were fabricated at varied ratios of graphene and served as co-photocatalysts for H$_2$ evolution. The results exhibited that the H$_2$ evolution of MTG-10 obtained is much better than others. The amount of hydrogen evolution was high, which was found to be 4122 μmol g$^{-1}$ of H$_2$ in 5 h with photocatalytic systems, which is almost 7.5~13.4 times greater than that of previous pristine MoS$_2$ (548 μmol g$^{-1}$) and TiO$_2$ (307 μmol g$^{-1}$) samples, respectively. This is significantly attributed to the graphene as a bridge of MoS$_2$/TiO$_2$ and the incorporation of graphene, suggesting the synergistic effect of the rapid electron-transferring of photoinduced electrons and holes and the powerful electron-collecting of graphene, suppressing the charge recombination rate.

**Keywords:** hydrogen production; photocatalytic nanohybrid; charge carriers separation; MoS$_2$/TiO$_2$/graphene

## 1. Introduction

The important environmental pollution and energy problem generated through fossil fuels has received a lot of attention recently. A photocatalytic activity system can employ reusable solar energy to create pure H$_2$ energy and cause environmental remediation [1], which is a favorable process in figuring out environmental and energy issues [2]. As a clean substitute energy, H$_2$ has attracted more attention because of its excellent energy content and environmental friendliness. Many studies have been performed on the improvement of better powerful semiconductor photocatalysts for H$_2$ evolution [3]. Owing to a suitable band gap structure and distinctive chemical and thermal stability, graphene is considered to be a good co-photocatalyst when it merges with other materials to form heterojunctions [4,5]. A heterostructure's photocatalytic is usually expressed as the operation in which reactions occur when a photocatalyst (like a semiconductor) absorbs enough solar energy to produce oxidative and reductive species, without the photocatalyst being consumed [6]. In the field of photocatalysts, the build-up of economical and useful photocatalytic samples for clean energy generation and environmental remediation has become one of the most significant issues. In particular, photocatalysts have issues of poor charge carrier separation rates and fast recombination efficiency. Now, solar H$_2$ serves as a potentially clean and renewable energy source of the future, owing to its good combustion capability, its high energy density, its non-toxicity combustion by-products, and the ease of switching to varied kinds of energy [7]. The promotion of capable photocatalysts has attracted expanded attention for their significant utilization in H$_2$ evolution through water splitting with visible light irradiation [8]. Previously, photoinduced electrons and hole pairs have been found to diffuse to the photocatalyst surface, where they contribute to the oxidation and

reduction reactions of water (H$_2$O) molecules [9,10]. During photocatalytic operation, the H$_2$O decomposition step is carried out via solar energy, which assists in the separation of charge carriers [11]. Therefore, these processes merge the benefits of photocatalytic and solar energy. Using photocatalysts in the form of nanohybrids further avoids the obstacles associated with the purification and separation of photocatalysts [12].

As a normal molybdenum disulfide, MoS$_2$, with a sandwich configuration of three stacked atomic layers (S-Mo-S), has been proven to be a superior photocatalyst towards H$_2$ evolution [13]. Pristine MoS$_2$, as a photocatalyst, has two existing faults: poor intrinsic conductivity and few active sites. Furthermore, titanium dioxide (TiO$_2$) has been the photocatalyst with the most potential due to it being not harmful and inexpensive and having a great photocatalytic capability and good chemical stability [14]. In addition, whereas TiO$_2$ prohibits band widths that are too vast and conducts quick charge carriers recombination [15], it is not enough for the efficient and quick evolution of H$_2$ in practical utilizations. Consequently, how to enhance the solar light harvest efficiency of TiO$_2$ and the separation performance of photo-response charge carriers is of interest to researchers. Recently, the carbon-based photocatalyst graphene, with excellent electron transport and reception capacity, has been broadly affected in the photocatalytic activity owing to the conjugated π structure of the carbon structure [16]. The introduction of graphene can improve the photocatalytic efficiency of photocatalysts and the application of visible light energy [17]. Graphene also has photosensitization, which can enhance electrons in the conduction band of the photocatalyst, improving the electron–hole density, forming valid electrons and a holes separation rate, and making graphene play a positive role in heterojunction photocatalysts [18,19]. Recently, several works about the suitable and useful photo-assisted deposition process for building an MoS$_2$/TiO$_2$ heterojunction have been published [20], illustrating the great photocatalytic efficiency of MB and 2-CP removal with visible light [21]. Hence, the main topic is to investigate high-performance nanohybrids with proper band gap energy and a low carriers recombination efficiency of photo-response electron–hole pairs.

In this work, the graphene-added MoS$_2$/TiO$_2$ heterojunction samples have been fabricated successfully for the first time via facile hydrothermal technology. In addition, the photocatalytic of the H$_2$ evolution efficiency of the synthesized MoS$_2$/TiO$_2$/graphene (MTG) nanohybrids was enhanced in relation to MoS$_2$ and TiO$_2$. The Z-scheme of MTG nanohybrids with tense interfaces enhanced the charge carrier separation capability and possessed the great reduction efficiency of photoinduced electrons. Finally, the MTG nanohybrids exhibited an excellent photocatalytic hydrogen production rate (HER), with light illumination and clean energy harvesting applications.

## 2. Results

### 2.1. Microstructure Characterization

X-ray diffraction pattern (XRD) tests were collected to check the crystal structure of the MoS$_2$, TiO$_2$, and MoS$_2$/TiO$_2$/graphene (MTG) nanohybrids. As displayed in Figure 1a, the diffracted peaks at 14.4°, 32.7°, 39.5°, 44.2°, 49.8°, and 60.2° corresponding to the (002), (100), (103), (006), (105), and (110) crystal plane for all samples, exhibit the MoS$_2$ with a hexagonal structure (PDF 73-1503) [22]. The secondary peaks at 25.3°, 37.7°, 47.9°, 53.9°, 55.1°, and 62.9°, corresponding to the (101), (004), (200), (105), (211), and (204) crystal plane in the MTG nanohybrids photocatalysts, were obtained for the anatase structure TiO$_2$ phase (PDF 21-1272) loading on the surface of MoS$_2$ as the first heterostructure (Figure 1a) [23]. As exhibited in Figure 1a, the diffraction peaks of the graphene were located at 25.5° of the XRD pattern [24]. Afterwards, the diffracted peaks did not appear in the XRD analysis of the MTG nanohybrids. This is caused by the dose of graphene loaded on the surface of the MoS$_2$/TiO$_2$ nanohybrid being lower than the detection limit of the XRD pattern. The UV–vis spectrum was studied for the MTG-2, MTG-5, MTG-10, and MTG-20 samples (Figure 1b). The absorbance of the MTG-20 nanohybrid, compared to that of MTG-2, MTG-5, and MTG-10, was significantly improved by adding the graphene. These optical

properties in the MTG-20 nanohybrid displayed more visible light absorbance than those of the MTG-2, MTG-5, and MTG-10, causing an improved dose of photoinduced electron and hole pairs while enhancing the charge carriers' mobility [25,26]. The optical absorption edge of the MTG-20 nanohybrid is 525 nm. The deposition of graphene as a co-photocatalyst with a narrow band gap onto $MoS_2/TiO_2$ significantly enhanced the absorption of solar light owing to the heterostructure development of graphene with $MoS_2/TiO_2$. Adding graphene also improved the absorption of solar light owing to the construction of a double heterostructure of graphene with $MoS_2$ and $TiO_2$. The band structures of the samples were estimated by the Kubelka–Munk (M-K) equation [27], $\alpha h v = A(h v - E_g)^{1/2}$, and acquired the corresponding band gap energy ($E_g$), as exhibited in Figure 1c, in which $\alpha$, $hv$, $A$, and $E_g$ are the absorption coefficient, photon energy, a constant, and the direct band gap (eV), respectively. The band gap values of the as-fabricated heterojunction that were calculated by the UV–vis absorption edges via the M-K equation exhibited that 2.41 eV of MTG-2 was decreased to 2.26 eV of MTG-20 after increasing with graphene. Therefore, it is indicated that MTG nanohybrid photocatalysts possess favorable band gap energy for absorbing solar light and approaching the transition from $H^+$ to $H_2$.

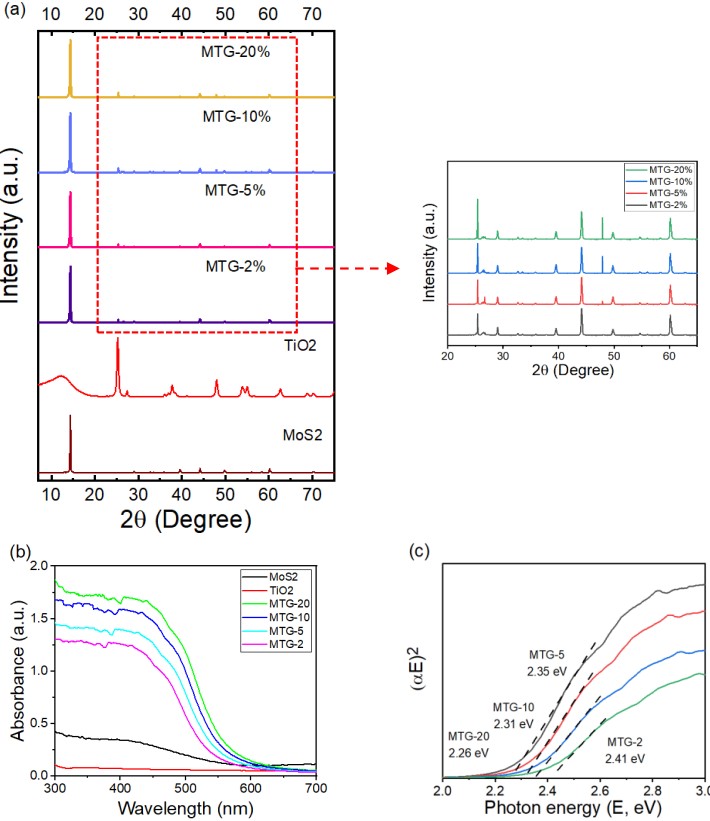

**Figure 1.** (**a**) X-ray diffraction test and (**b**) UV–vis absorbance of the samples; (**c**) Band gap plot of as-synthesized MTG nanohybrid photocatalysts with different amounts of graphene.

The morphology of fabricated photocatalysts was received via SEM tests. As displayed in Figure 2a, the $MoS_2$ photocatalyst shows large sheets with a relatively flat surface and an average dimension of several micrometers. The morphology of $TiO_2$ photocatalysts has developed particles with an average dimension of 2~3 μm, which have been contained within the spheres in Figure 2b. The morphology of the MTG-10 nanohybrid exhibits only alike nanoparticles in Figure 2c. Hence, the obtained photocatalyst in the MTG-10 nanohybrid structure could be referred to the $MoS_2$ and $TiO_2$. TEM images are recorded to study the specific structure of the fabricated nano-scale structures and their composites. Figure 2d displays the TEM tests of the MTG-10 nanohybrid. Fabricated MTG-10 nanohybrids are uniform in the sizes of the nanoparticles (~15–18 nm). Figure 2e exhibits



a histogram of the diameter of nanohybrids received from the TEM microscopy test. The samples distribution histogram was achieved via 200 samples from the TEM micrograph. The results referred to the log–normal distribution fit with an average particle size of 17 nm and a standard deviation of 0.24. The elemental composition distribution of MTG-10 nanohybrids was studied through energy dispersive spectra (EDS) element mapping tests in Figure 2f–k. The results implied the presence of Mo, S, Ti, O, and C elements in the MTG-10 nanohybrids, and it is apparent that the elements were distributed uniformly. The height coincidence of elemental distribution could be obtained in the Mo element and Ti element, which verifies the preparation of MTG nanohybrids. Then, the height coincidence was observed for the S element and the C element, mostly referring to the graphene added. Despite the pronounced height coincidence, however, an evident diverse area could be noted through comparing the S element with the C element, which was certainly consistent with the distribution distinction between the Mo element and the Ti element at the same location. Furthermore, the well-fabricated MTG nanohybrids were regularly anchored all over the heterojunction interface and onto the surface of the nanohybrid itself, establishing the construction of the nanohybrid interface [28]. The Fourier Transform Infrared (FT-IR) curves of the MoS$_2$, TiO$_2$, and MTG-10 samples are characterized to check the interaction in the heterostructures in Figure 2l. The total patterns of samples are similar, suggesting that the presence of graphene formation in the MTG-10 samples does not significantly affect the basis construction of the MoS$_2$ and TiO$_2$. The feature peak of the aromatic ring plane vibration could be noted from 1150 cm$^{-1}$ to 1800 cm$^{-1}$.

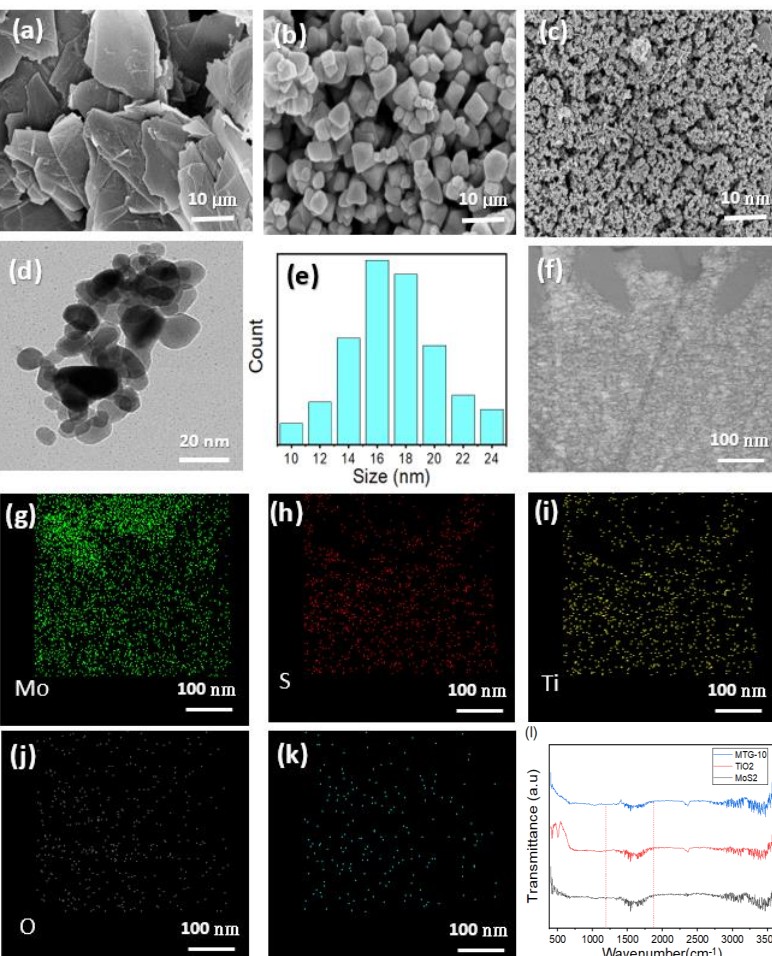

**Figure 2.** SEM images of (**a**) MoS$_2$, (**b**)TiO$_2$, (**c**) MTG−10; TEM images of (**d**) MTG−10; (**e**) average particle size obtained by the histogram image; element mapping images (**f–k**) of MTG−10; (**l**) FT−IR of samples.

We used the X-ray photoelectron spectroscopy (XPS) to study the structure and surface element composition of different catalysts in Figure 3. The Mo 3d, S 2p, Ti 2p, O 1s, and C 1s binding energies were related to the C 1s peak at 284.9 eV. From the survey spectra in Figure 3a, the binding energies of the electrons of the Mo, S, Ti, O, and C elements could be obtained. The feature peaks assigned at 229.46 eV and 232.65 eV correspond to the Mo 3d5/2 and Mo 3d3/2 [29], respectively, as exhibited in the high-resolution XPS spectrum of Mo 3d in Figure 3b. In addition, Figure 3c reveals the deconvolution of the high-resolution S 2p spectra in the nanohybrid. It displayed two major feature peaks at 162.37 and 163.49 eV, referring to S 2p 3/2 and S 2p 1/2, respectively [29]. Furthermore, the XPS spectrum of Ti represents two peaks at 459.68 and 464.95 eV, referring to Ti 2p 3/2 and Ti 2p 1/2 of the MTG-10 sample (Figure 3d), respectively [30]. The Gaussian peak fitting of O 1s in the nanohybrid exhibited three varied peaks at 530.98, 531.92, and 532.76 eV (Figure 3e) and adsorbed oxygenated species, respectively. The high-resolution spectra of C 1s in the MTG-10 photocatalyst (Figure 3f) reveal three feature peaks at 284.18, 284.95, and 285.61 eV, respectively [31].

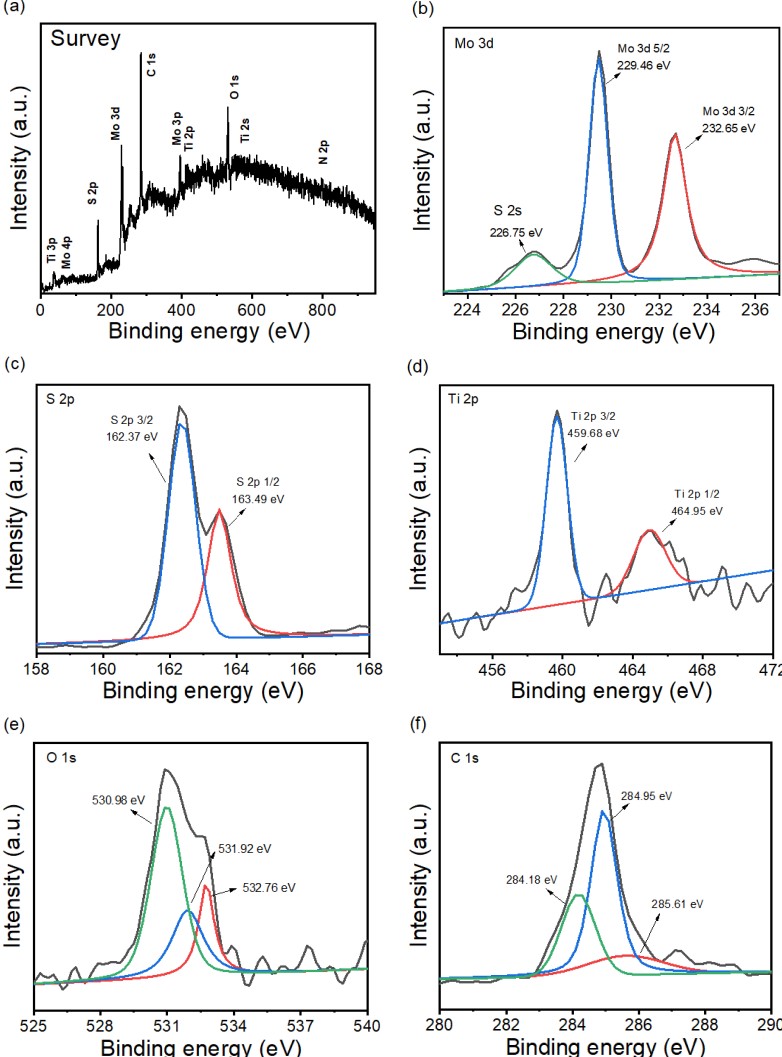

**Figure 3.** XPS patterns of MTG-10. (**a**) Survey XPS spectrum; (**b–f**) The high-resolution XPS spectrum of Mo2p, S2p, Ti2p, O 1s, and C1s.

### 2.2. Photoelectrochemical Behavior

The separation and transport of photo-response charge carriers in photocatalysts are of intense significance [32]. Mainly, photocatalysts with better photocatalytic efficiency are

normally practical for the valid transfer and separation of photo-response charge carriers. Primarily, photoluminescence spectrum (PL) and EIS tests were performed to explore the separation and recombination behavior of photo-response electrons in semiconductors. The peak intensity in the PL spectrum reflects the recombination capability of photo-response electrons and holes. As expected, the PL intensities order was MTG-2 > MTG-5 > MTG-20 > MTG-10, implying that the recombination ability of electron–hole pairs diminished gradually. The lesser the PL intensity, the lesser the recombination rate and, therefore, the greater the separation performance [32,33]. Figure 4a shows that the primary PL feature peak for MTG-10 is located at nearly 500 nm, which refers to the transition between its states of the band gap and reveals an excellent electron–hole recombination efficiency. Compared with MTG-2, MTG-5, and MTG-20, the PL feature peak intensity of MTG-10 decreases obviously. Especially, the PL feature peak intensity of MTG-10 is the lowest. These PL spectrum analyses verify that the MTG-10 nanohybrid can practically transfer photo-response charge carriers to inhibit their recombination rate. The transportation of photo-response charge carriers has been studied via electrochemical impedance spectroscopy (EIS). In Figure 4b, the Nyquist semicircle radius of the MTG-10 nanohybrid is obviously smaller than that of MTG-2, MTG-5, and MTG-20, suggesting that the transmission resistance of photo-response charge carriers in the MTG-10 sample is the smallest among these photocatalysts. The results demonstrate that the formed nanohybrid can enhance transport photo-response electron–hole pairs, which is very efficient in promoting the photocatalytic efficiency of the photocatalysts [34,35]. The photocatalytic efficiency of the MTG nanohybrid was studied via measuring the photocurrent responses at 0 V in Figure 4c [36]. It is clear that the jointing of graphene to $MoS_2/TiO_2$ improved the photocatalytic efficiency, and the MTG nanohybrid exhibited the best photocurrent density value of 37.5 μA cm$^{-2}$ compared to the others. This suggests its improved photocatalytic capability. Otherwise, MTG-2 displayed an insignificant photocurrent signal, which confirms its lower photocatalytic activity in Figure 4c. As for the transient photocurrent response tests results, the photocurrent intensities of the as-fabricated samples exhibited the order of MTG-2 <MTG-5 < MTG-20 < MTG-10. The photocurrent density value of MTG-10 is 3.24-fold that of the MTG-2 sample, 2.35-fold that of the MTG-5 sample, and 1.04-fold that of the MTG-20 sample. Stability is a critical factor for using a material as a photocatalyst. Figure 4d describes the variety of photocurrent responses of the MTG-10 nanohybrid with the visible light irradiation time. Particularly, the photocurrent density exhibited no apparent decrease from its initial value after irradiation with visible light for 800 s. This suggests the improved photostability of the nanohybrid. All of the above results obviously verify that the photo-response charge carriers in the MTG-10 nanohybrid could be successfully transferred and separated.

　　To research the active radicals under the photocatalytic reaction system, the electron paramagnetic resonate (EPR) spectrum of the MTG-10 nanohybrids was recorded with visible light with various illumination times, and the results are shown in Figure 5. No feature signal could be obtained with dark conditions. During the visible light response, four sets of antisymmetric feature signals with intensities of 1:2:2:1 of MTG-10 nanohybrids could be received. The four signals are the feature peaks of the DMPO-•OH adducts. With an extended irradiation time, the feature peak intensities of MTG-10 nanohybrids display an insignificant increase, which suggests that the number of •$O_2^-$ radicals generated has enhanced [37]. The EPR results indicate that •OH radicals are the major active species of the MTG-10 nanohybrid in the photooxidation activity with the visible light response. The feature peak intensity of the MTG-10 nanohybrid is much stronger than that of the sample with the increase in irradiation time, which demonstrates that more •OH radicals could be generated with 60 s than with 10 s under the same conditions, and this is assigned to the photoinduced electron migration performance of the as-fabricated MTG-10 nanohybrid. The feature peak of DMPO–•$O_2^-$ can roughly be received, which indicates that •OH is the dominant active radical of this photocatalytic system. With the prolongation of the visible light irradiation time, the signal intensities of EPR gradually enhance, while the signal intensities of •OH gradually increase, suggesting that the MTG sample can oxidize

water ($H_2O$) and also convert it to $^\bullet OH$ radicals. Owing to the better redox efficiency of $^\bullet O_2{}^-$ radicals and $^\bullet OH$ radicals, $H_2O$ decomposition and $H_2$ production could be achieved. Based on the above results, a nanohybrid is developed between graphene and $MoS_2/TiO_2$ in the MTG-10 sample. The addition of graphene can improve the photo-response charge carriers' transfer ability in the heterostructure and extend the photo-response electron lifetime to enhance the separation performance of the photoinduced electron–hole pairs [38]. Graphene in the MTG-10 heterojunction forms a stable Z-type system, which enhances the visible light operation performance [39].

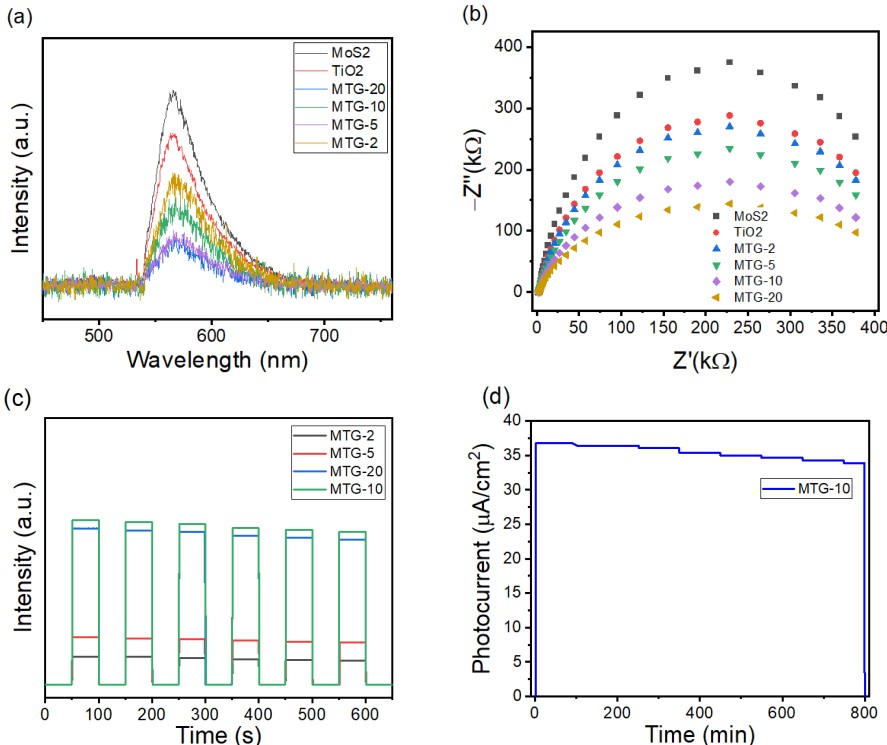

**Figure 4.** (**a**) Photoluminescence (PL) spectra, (**b**) electrochemical impedance spectroscopy (EIS), (**c**) transient photocurrent response, and (**d**) the photocurrent stability of the as-fabricated photocatalysts.

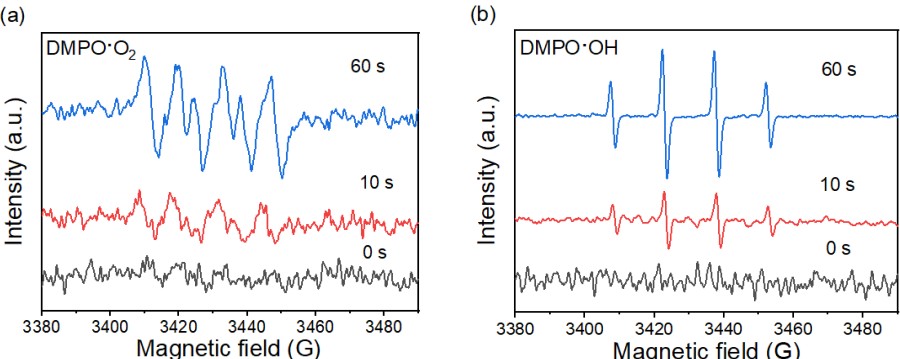

**Figure 5.** EPR spectrum of the radical adducts DMPO-$^\bullet O_2{}^-$ (**a**) and DMPO-$^\bullet OH$ (**b**) in MTG-10 nanohybrids under dark and visible light activity with different times at room temperature.

### 2.3. Hydrogen ($H_2$) Evolution Behavior

The photocatalytic $H_2$ evolution effect of $MoS_2$, $TiO_2$, and MTG nanohybrids was performed with visible light, with triethanolamine (TEOA) as the sacrificial reagent. As exhibited in Figure 6a, the $H_2$ production efficiency of MTG-10 improves by 7.6 and 13.5 times compared to those of the $MoS_2$ and $TiO_2$ samples, respectively. Furthermore,

the trend in the $H_2$ evolution efficiency of the MTG nanohybrids exhibits a "sharp" form, implying that graphene influences the photocatalytic capability of the MTG nanohybrids (Figure 6b). When the addition of graphene was excessive, the graphene aggregated on the surface of the sphere-like MTG nanohybrids, suggesting an increase in the surface ruggedness of MTG nanohybrids and hence prohibiting the internal diffusion of the sacrificial reagent [40]. Excessive graphene may develop recombination centers for photoinduced charge carriers and then decrease the photocatalytic activity. Further, a slight amount of graphene will produce the interface between the $MoS_2$ and $TiO_2$, limited to migrating and separating the photo-response electron–hole pairs, which cannot efficiently suppress the recombination rate of photo-response charge carriers. Afterward, we studied the performance of MTG-10 in relation to TEOA at various pH values. The $H_2$ production in the TEOA solution at pH = 7, 8, 9, 10, 11, and 12 is exhibited in Figure 6c. As the solution alters from pH 7 to pH 12, the efficiency of $H_2$ production further presents a "sharp" mode. The $H_2$ production is the best at pH 9. This is caused by the diminished concentration of $H^+$ in the solution being negative to $H_2$ production in terms of thermodynamic kinetics and the electrostatic repulsion between the reaction solution and graphene added [41]. In the photocatalytic activity, graphene is adsorbed on the surface of the nanohybrids so that this graphene can contribute to the photocatalytic system. Therefore, it is required to explore the optimal amount of the photocatalyst included in the photocatalytic process. As exhibited in Figure 6d, with the addition of photocatalysts, the efficiency of $H_2$ production enhances gradually until the amount of the photocatalyst approaches 40 mg. After that, the $H_2$ production reaction is gradually suppressed with an increasing photocatalyst amount. This suggests that it is usefully applied when the amount of the photocatalyst is suitable. Once the amount of the photocatalyst exceeds 40 mg, excess photocatalyst collects in the system, forbidding the photon adsorption and practical electron migration of the sample and then reducing the number of photons on the surface of the sample, implying reduced $H_2$ evolution operation [42]. To investigate the stability of photocatalysts, an $H_2$ production cycle experiment was achieved. As exhibited in Figure 6e, the efficiency of $H_2$ evolution in the fifth cycle test was slightly decreased compared with that in the first cycle. Figure 6f shows when the sacrificial agent was methanol and ethanol, the hydrogen production activity significantly decreased, with methanol resulting in the highest hydrogen production activity of only 384.3 $\mu$mol $g^{-1}$ $h^{-1}$, indicating that these alcohols had inferior hole-capture abilities compared with TEOA. The $H_2$ evolution rate over the MTG-10 photocatalyst in methanol, ethanol, and TEOA aqueous solutions were 384.3, 327.6, and 824.6 $\mu$mol $g^{-1}$ $h^{-1}$, respectively. The $H_2$ production performance of MTG-10 with 3.0 wt.% and 1.0 wt.% Pt as cocatalysts was 382.3 and 484.5 $\mu$mol $g^{-1}$ $h^{-1}$, as shown in Figure 6g. For MTG-10 with 1.0 wt.% Pt, the $H_2$ production performance was better than that of MTG-10 with 3.0 wt.% Pt as a cocatalyst. In addition, the $H_2$ production performance of MTG-10 without a Pt of 824.6 $\mu$mol $g^{-1}$ $h^{-1}$ was approached. It is implied that MTG-10 indicates the optimal efficiency, and the suggested $H_2$ production performance of MTG-10 is 2.2- and 1.7-fold better than that of MTG-10 with 3.0 wt.% Pt and MTG-10 with 1.0 wt.% Pt, respectively. At the end (fifth) of the cycle experiment, the XRD analysis of the nanohybrid was performed again. As exhibited in Figure 6h, the positions of the XRD feature signals before and after cycling do not obviously alter, suggesting that the photocatalyst has superior cycle stability during this system.

According to the abovementioned results, a possible photocatalytic $H_2$ production mechanism can be proposed (Figure 7). During the visible light activity, photoexcited electrons in both $TiO_2$ and $MoS_2$ are induced from VB to their CB, respectively, with holes moved on their VB. In addition, donated through the vigorous interface conjugation between $TiO_2$ and $MoS_2$, the electrons transport from the CB of $TiO_2$ to the VB of $MoS_2$ and suppress the photoexcited holes because of a Z-scheme charge carriers transport process. Additionally, owing to the low Fermi level of graphene, the formed Schottky junction between $MoS_2$ and $TiO_2$ could further accelerate the charge transfer and inhibit the electron–hole recombination rate. Consequently, these photoinduced charges can be

transferred to the surface and conduct the $H_2$ production more successfully. The interfacial graphene, which acts as an interfacial bridge, can remarkably accelerate the separation of photoexcited electrons and holes, resulting in a higher charge transfer efficiency between $TiO_2$ and $MoS_2$. In this work, we indicated that graphene, as a conductive "bridge", can improve the valid charge carriers separation efficiency and be a useful interfacial charge transfer path [43]. This result provides a basis for the realization of the design of photocatalysts for $H_2$ production and for enhancing the stability of the heterojunction to photocorrosion.

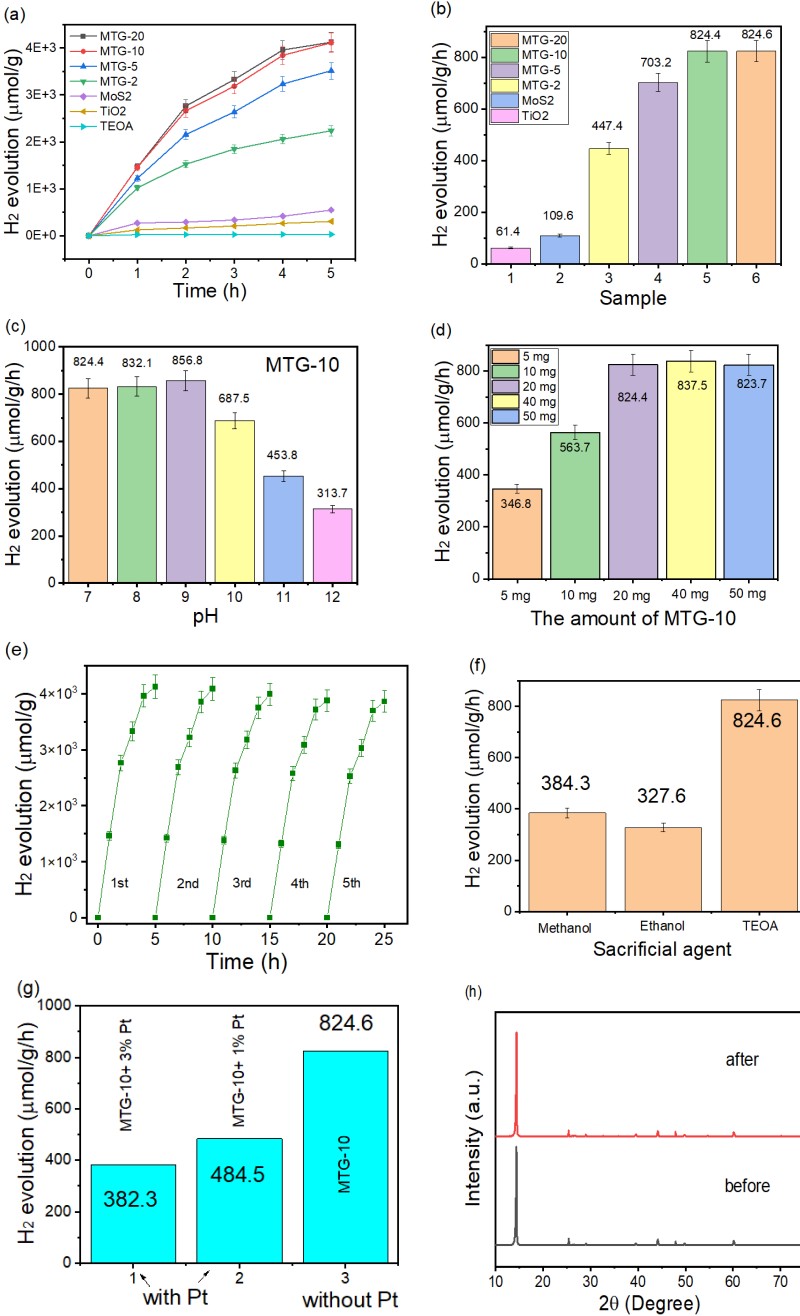

**Figure 6.** $H_2$ production of (**a**,**b**) $MoS_2$, $TiO_2$, and MTG nanohybrids (MTG-2, MTG-5, MTG-10, and MTG-20). (**c**) $H_2$ production of MTG-10 in TEOA with various pH values in three repeated tests. (**d**) $H_2$ production with various amounts of MTG-10. (**e**) $H_2$ production of the cycling experiment. (**f**) Photocatalytic $H_2$ production activity over different sacrificial agents. (**g**) $H_2$ evolution efficiency of MTG-10 with and without Pt. (**h**) XRD analysis of MTG -10 after and before the $H_2$ production test.

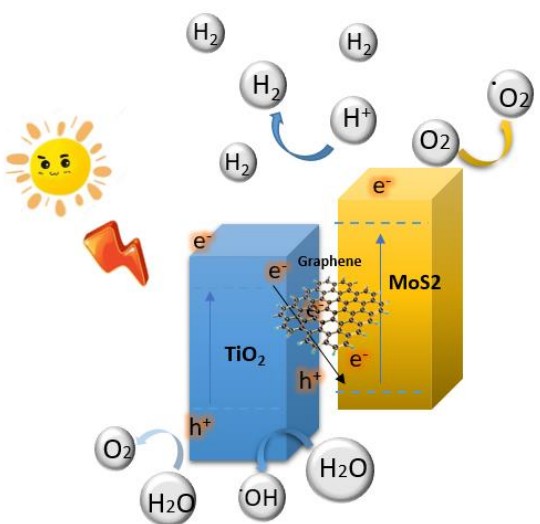

**Figure 7.** Illustration of the charge transfer and $H_2$ precipitation process in MTG nanohybrids under light irradiation.

## 3. Materials and Methods

### 3.1. Preparation of Photocatalysts

All reagents were purchased from Sigma Aldrich (St. Louis, MO, USA), with no subsequent decontamination. Sodium dihydromyldimolate ($Na_2MoO_4$ $2H_2O$), L-cysteine ($C_3H_7NO_2S$), titanium butoxide (Ti(OBu)$_4$, 98%), 2-propanol, tetrabutylammonium hydroxide (TBA), and graphene were supplied from Sigma Aldrich.

### 3.2. Synthesis

The fabrication of nanohybrids and the hydrothermal process are illustrated in Scheme 1. Briefly, 1.2 mmol of $Na_2MoO_4$ $2H_2O$ was mixed with 20 mL of DI water, with vigorous stirring for 40 min (Precursor A solution). $TiO_2$ nanoparticles were fabricated, adopting the hydrothermal route [44]. In this work, a 0.5 M titanium butoxide (Ti(OBu)$_4$) solution in 2-propanol was mixed in DI water, with magnetic stirring for 50 min (Precursor B solution). The received residue was filtered to obtain the $TiO_2$ product. The $TiO_2$ product was added in tetrabutylammonium hydroxide (TBA) and dilute $HNO_3$ solutions and heated at 80 °C for 30 min with magnetic stirring. Subsequently, 2.4 mmol of L-cysteine and 2.2 mmol of $TiO_2$ were delivered to 30 mL of DI water for 30 min, with magnetic stirring (Precursor C solution). Also, HCl (0.1 mL) and varied dosages (2, 5, 10, and 20 mg) of graphene were removed from the abovementioned two solutions with fixed stirring to obtain various ratios of $MoS_2/TiO_2/graphene$ (MTG) (Precursor D solution). Next, the mixed suspension was transferred to a 50 mL Teflon-lined stainless-steel autoclave for hydrothermal treatment at 180 °C for 16 h. The obtained products were expressed as MTG-2, MTG-5, MTG-10, and MTG-20, respectively, based on the sum of graphene.

### 3.3. Characterization

The crystal structure of the samples was studied by X-ray diffraction (XRD) Bruker D8 Advanced (Billerica, MA, USA) with Cu Kα radiation. The microstructures and morphologies of the samples were checked by a scanning electron microscope (SEM, ZEISS AURIGA, Oberkochen, Germany) and transmission electron microscope (TEM, JEOL JEM-2100F, Tokyo, Japan). Electron paramagnetic resonance (EPR, Munich, Germany) was used as the trapping agent to trap the hydroxyl radical ($^\cdot$OH) or the superoxide radical ($\cdot O_2^-$) on Bruker A300 under visible light. The UV–vis absorption spectra were recorded by a UV–vis spectrometer (Hitachi UV-4100, Tokyo, Japan). X-ray photoelectron spectroscopy (XPS) was used on a Thermo ESCALAB spectrometer (XPS, Thermo Fisher Scientific, Waltham, MA, USA) with an Al Kα source.

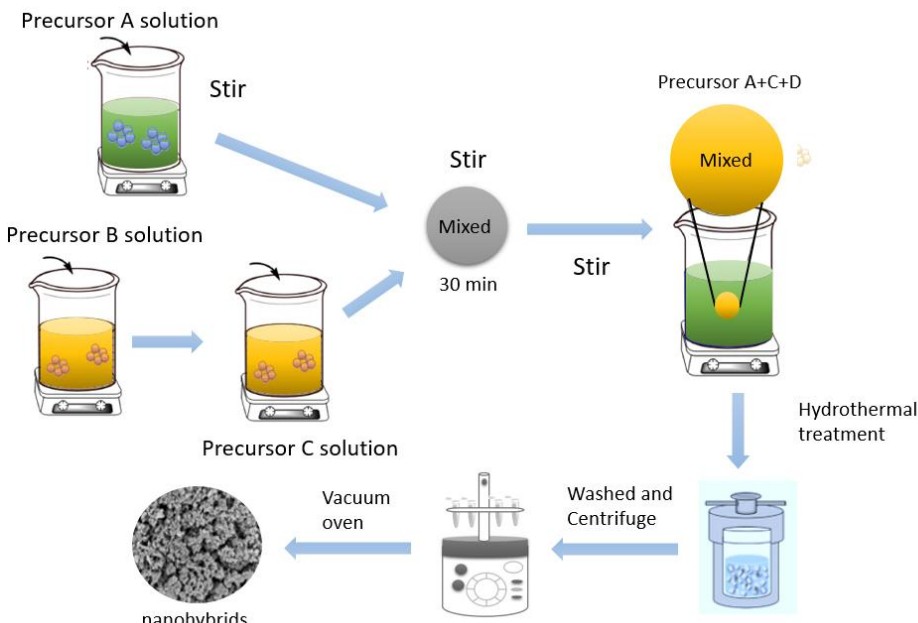

**Scheme 1.** The schematic illustration of the synthesis process of photocatalysts.

### 3.4. Photocatalytic Hydrogen Evolution Tests

The photocatalytic $H_2$-generated activity of the as-obtained nanohybrid was further measured by GC 7890 gas chromatography with a TCD detector from an aqueous solution. The types and concentrations of the sacrificial agent are 10 vol% triethanolamine (TEOA) and the equimolar amount of organic alcohol, including methanol and ethanol. Mostly, the photocatalytic $H_2$ production reactions of the nanohybrids were achieved in a photocatalytic system composed of a double-walled quartz reaction vessel combined with a close gas rotation. Subsequently, a Xe lamp of 350 W with simulated solar irradiation AM 1.5 G (100 mW/cm$^2$) was used as a light source. Visible light (410 < $\lambda$ < 790 nm) was used as an irradiation source, which was achieved by using both a 790 nm cutoff filter (short-wave-pass) and a 410 nm cutoff filter (longwave-pass). In all cases, the reaction temperature was contained at 12 °C by flowing cold water, adopting a flowing water jack. A total of 20 mg of the sample was spread in a mixture of 100 mL of water and 10 mL of methanol. During this photocatalytic activity, argon gas as the carrier gas was used through the reaction merger under the dark to eliminate oxygen and ensure the anaerobic status. After the removal of air with a highly pure argon gas, the light was turned on for 60 min without photocatalysts. It is properly stated that the reaction vessel was not vacuumed. Stability tests were performed under the same situation.

### 3.5. Photoelectrochemical (PEC) Analysis

The photoelectrochemical (PEC) tests of the samples were recorded by PEC worksta-tions, consisting of electrochemical impedance spectroscopy (EIS) and a transient photocur-rent response. The PEC measurement was analyzed on a three-electrode electrochemical workstation, where the reference electrode was a saturated calomel electrode, a Pt electrode was the counter electrode, and the working electrode was composed of a conductive glass substrate (FTO) coated with photocatalysts. The preparation process for the working elec-trode is as follows: 10 mg of the sample was added in a mixture solution of 1 mL of alcohol and 0.1 mL of Nafion. The mixture solution was then coated on an FTO substrate with an effective area of 1.5 × 1.5 cm$^2$ and dried for 10 h in the air. EIS was recorded at a voltage extent of 10 mV. The photocurrent response was performed within 600 s under a 350 W xenon lamp as the visible light source, and the wavelength was arranged to above 420 nm via a filter.

## 4. Conclusions

In summary, the $MoS_2/TiO_2$/graphene (MTG) nanohybrids with heterojunction interfacial contact were built via a facile hydrothermal process. We established the visible light irradiation activity and the higher contribution to the $H_2$ production of MTG nanohybrids. The charge carriers with high-energy photon excitation can be utilized via the surface defects of the MTG nanohybrid, which is achieved as the photocatalytic central and accelerates the $H_2$ production process. Therefore, the $H_2$ production efficiency of the MTG nanohybrids was obviously enhanced to 824.6 $\mu$mol g$^{-1}$ h$^{-1}$ with visible light irradiation, which was 13.5-fold and 7.6-fold greater than that of pristine $TiO_2$ (61.4 $\mu$mol g$^{-1}$ h$^{-1}$) and $MoS_2$ (109.6 $\mu$mol g$^{-1}$ h$^{-1}$), respectively. The matched energy band structure between graphene and $MoS_2/TiO_2$ enhances the separation ability of photo-induced charge carriers and restrains the recombination of hole–electron pairs. This is significantly attributed to the graphene as a bridge of $MoS_2/TiO_2$ and the incorporation of graphene, suggesting the synergistic effect of the rapid electron-transferring of photoinduced electrons and holes and the powerful electron-collecting of graphene, suppressing the charge recombination rate. The current work not only provides insight into a graphene-bridged heterojunction photocatalytic for $H_2$ evolution but also motivates people to actively explore novel heterojunctions with various photocatalysts towards various applications.

**Author Contributions:** Writing—original draft, Methodology, T.-M.T.; Writing—review and editing, E.L.C.; Data curation, Formal analysis, T.-M.T.; Funding acquisition, T.-M.T.; Writing— review and editing, T.-M.T.; Validation, E.L.C.; Resources, T.-M.T.; Formal analysis, Data collection, E.L.C.; Conceptualization, Writing—review and editing, E.L.C.; Supervision, Investigation, E.L.C.; Supervision, Project administration, T.-M.T. All authors have read and agreed to the published version of the manuscript.

**Funding:** This research received no external funding.

**Data Availability Statement:** The data presented in this study are available on request from the corresponding author.

**Acknowledgments:** The MOST and the NKUST are gratefully acknowledged for their general support. The authors gratefully acknowledge the use of HRTEM equipment belonging to the Instrument Center of National Cheng Kung University.

**Conflicts of Interest:** The authors declare no conflict of interest.

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
