# Peer review of "A Novel MoS2/TiO2/Graphene Nanohybrid for Enhanced Photocatalytic Hydrogen Evolution under Visible Light Irradiation"

_catalysts, doi:10.3390/catal13081152_

Round 1
Reviewer 1 Report
The article is devoted to the development and study of the photocatalytic properties of a new material based on molybdenum disulfide, ttanium oxide and graphene. The conducted researches are objective and meaningful. The article can be published after correcting the following comments:
1. What are the units of measurement for the Y axes in Figures 6 d,b?
2. According to figures 2 c, e, the particle size is from 10 to 20 nm. What is shown in figure 2 f with a scale of 20 nm?
3. What is the filamentous structure of molybdenum disulfide in Figures 2 f g h?
4. Why are gray and white structures clearly visible in the 2 f image, but there are no elements in the elemental analysis of the white areas?
5. Change the color of the carbon in Figure 2j. The image is hard to see
6. Which device allows elemental mapping on an area of about 100 nm2 according to Figures 2 f-j?
7. Why is oxygen not visible from titanium oxide according to the text and figure 2 f-j?
The results implied the presence of Mo, S, Ti, and C elements in the MTG-10 (line 172)
8. In figure 1 a - the peaks are difficult to distinguish
9. At what wavelength do the authors believe that visible light begins? According to https://en.wikipedia.org/wiki/Ultraviolet#NUV, the ultraviolet range of electromagnetic waves ends at 400 nm. Why is the UV-visible absorption range stated on the caption (Fig. 1b), while the graph shows absorption from 400 nm?
Author Response
Author's Reply to the Review Report (Reviewer 1)
Reviewer1
Comments and Suggestions for Authors
The article is devoted to the development and study of the photocatalytic properties of a new material based on molybdenum disulfide, ttanium oxide and graphene. The conducted researches are objective and meaningful. The article can be published after correcting the following comments:
- What are the units of measurement for the Y axes in Figures 6 d,b?
Reply: Thank you for the correction. Photocatalytic H2 evolution efficiencies depend strongly on the lifetime of the excited state and transport of the photo-generated charge carriers. Various strategies have been explored to enhance charge transport, such as tuning surface facets, loading cocatalysts, and constructing heterojunctions [r1]. The evolution of H2 by photocatalyst has been measured by GC 7890 gas chromatography with a TCD detector. H2 evolution rates of samples (Fig.6b and Fig. 6d) for 61.4~837.5 µmol g-1 h-1 are achieved respectively with this system (although H2 evolution rates on the order of µmol g-1 h-1 are needed for practical applications) [r2].
r1 Chen, R.; Fan, F.; Dittrich, T.; Li, C. Imaging photogenerated charge carriers on surfaces and interfaces of photocatalysts with surface photovoltage microscopy. Chem. Soc. Rev. 2018, 47, 8238–8262.
r2 She, X.; Wu, J.; Xu, H.; Zhong, J.; Wang, Y.; Song, Y.; Nie, K.; Liu, Y.; Yang, Y.; Rodrigues, M. F.; Vajtai, R.; Lou, J.; Du, D.; Li, H.; Ajayan, P. M. High efficiency photocatalytic water splitting using 2D α-Fe2O3/g-C3N4 Z-scheme catalysts. Adv Energy Mater. 2017, 7, 1700025.
- According to figures 2 c, e, the particle size is from 10 to 20 nm. What is shown in figure 2 f with a scale of 20 nm? 3. What is the filamentous structure of molybdenum disulfide in Figures 2 f g h? 4. Why are gray and white structures clearly visible in the 2 f image, but there are no elements in the elemental analysis of the white areas? 5. Change the color of the carbon in Figure 2j. The image is hard to see 6. Which device allows elemental mapping on an area of about 100 nm2according to Figures 2 f-j? 7. Why is oxygen not visible from titanium oxide according to the text and figure 2 f-j? The results implied the presence of Mo, S, Ti, and C elements in the MTG-10 (line 172)
Reply: Special thanks to you for your good comments. It’s true that our original analysis is not clearly enough. According to your suggestion, we have modified Fig. 2 f~k and shown it in the revised manuscript. It has been described in Page 6, Lines 219~222: “ Besides, the elemental composition distribution of MTG-10 nanohybrids was studied through energy dispersive spectra (EDS) element mapping tests in Fig. 2f-k. The results implied the presence of Mo, S, Ti, O, and C elements in the MTG-10 nanohybrids, and it is apparent that the elements were distributed uniformly. ”
- In figure 1 a - the peaks are difficult to distinguish
Reply: Thanks for the kind suggestion. We have modified Figure 1a and shown it in the revised manuscript.
Fig. 1. (a) X-ray diffraction test of the samples.
- At what wavelength do the authors believe that visible light begins? According to https://en.wikipedia.org/wiki/Ultraviolet#NUV, the ultraviolet range of electromagnetic waves ends at 400 nm. Why is the UV-visible absorption range stated on the caption (Fig. 1b), while the graph shows absorption from 400 nm?
Reply: Thanks for the kind suggestion. Although the UV–Vis DRS results of the MTG-2, MTG-5, MTG-10 and MTG-20 samples demonstrated typical band gap absorption, the MTG sample exhibited wideband absorption from 300 to 650 nm. Thus the graphene extends the light absorption of MTG sample to visible region [r3]. The absorption range of MTG-20 showed an increased intensity of visible-light absorption, which is due to the strong optical absorption of graphene[r4-r6]. We have modified Figure 1 and shown in the revised manuscript.
Fig. 1. (b) UV- vis absorbance of the samples.
r3 Lavorato, C.; Primo, A.; Molinari, R.; Garcia, H. N-doped graphene derived from biomass as a visible-light photocatalyst for hydrogen generation from water/methanol mixtures, Chem. Eur. J. 2014, 20, 187.
r4 Shaikh, Z. A.; Laghari, A. A.; Litvishko, O.; Litvishko, V.; Kalmykova, T.; Meynkhard, A. Liquid-Phase Deposition Synthesis of ZIF-67-Derived Synthesis of Co3O4@TiO2 Composite for Efficient Electrochemical Water Splitting, Metals 2021, 11, 420.
r5 Li, X.Y.; Shao, J.; Li, J.; Zhang, L.; Qu, Q.T.; Zheng, H.H. Ordered mesoporous MoO2 as a high-performance anode material for aqueous supercapacitors. J. Power Sour. 2013, 237, 80–83.
r6 Ghodke, N.P.; Rayaprol, S.; Bhoraskar, S.V.; Mathe, V.L. Catalytic hydrolysis of sodium borohydride solution for hydrogen production using thermal plasma synthesized nickel nanoparticles. Int. J. Hydrog. Energy 2020, 45, 16591–16605.

Reviewer 2 Report
1. Capitalize the first alphabet of each word in the title and keywords section and so one.
2. Giving a clear comparison of your work with your previous reported work in the abstract and conclusion section, show the novelty specially.
3. Show numerical values of your photocatalytic performance and its comparison enhancement in the abstract section and conclusion.
4. Give a descriptive schematic mechanism for the photocatalytic hydrogen evolution along with reaction step.
5. Give a separate section in the text to discuss the photocatalytic hydrogen evolution mechanism in detail with references. This topic will be probably at the top portion of conclusion section.
6. In figure 6c and d, write the names of materials instead of digits.
7. You do not show the light. You used visible light or Uv-light. If you used visible light, then which filter cut you used. I do not see any information related. Repeat the whole paper properly.
8. Show the effect of different sacrificial agent with your results.
9. Provide the important analysis such as surface area of all materials probably in form of table, photoluminescence spectra (PL) and DRS of materials. These parameters are important for one photocatalyst.
10. Provide the FTIR analysis.
11. Why you used 10 ml methanol with 100 ml of water. What is the role of methanol even you used the sacrificial agent TEOA in photocatalytic water splitting. Explain with facts.
12. You detect the photocatalytic hydrogen evolution without the induction of co-catalyst?? In case show the effect of co-catalyst as comparison. In my opinion, such performance is not possible without co-catalysts specially Pt. All the materials you used have large bandgap and almost unactive under visible light. It is better that you provide the following parameters
(a) Band gap of materials
(b) Effect of co-catalyst in comparison with no co-catalyst performance.
13. Double check the English corrections, grammar checks, pronunciation, punctuation, and other essential changes in the revised manuscript copy.
14. Some papers in the introduction portion need to be cited for good interest
https://doi.org/10.1016/j.mattod.2023.02.025
https://doi.org/10.1016/j.ensm.2023.102780
Improved
Author Response
Reviewer2
Comments and Suggestions for Authors
- Capitalize the first alphabet of each word in the title and keywords section and so one.
Reply: Thanks for the kind suggestion. We have been revised and modified as follow above.
- Giving a clear comparison of your work with your previous reported work in the abstract and conclusion section, show the novelty specially.
- Show numerical values of your photocatalytic performance and its comparison enhancement in the abstract section and conclusion.
Reply: Thanks for the kind suggestion. It has been described in Abstract and Conclusion: “ The amount of hydrogen evolution was high which was found to be 4122 μmol g-1 of H2 in 5 h with photocatalytic systems, which is almost 7.5~13.4 times greater than that of previous pristine MoS2 (548 μmol g-1) and TiO2 (307 μmol g-1) samples, respectively. This is significantly attributed to the graphene as a bridge of MoS2/TiO2 and the incorporation of graphene suggesting the synergistic effect of rapid electron-transferring of photoinduced electrons and holes, and the powerful electron-collecting of graphene suppressing charge recombination rate.” And “ Therefore, the H2 production efficiency of MTG nanohybrids was obviously enhanced to 824.6 μmol-g-1-h-1 with visible light irradiation, which was 13.5 folds and 7.6 folds greater than that of pristine TiO2 (61.4 μmol-g-1-h-1) and MoS2 (109.6 μmol-g-1-h-1), respectively. The matched energy band structure between graphene and MoS2/TiO2, which enhances the separation ability of photo-induced charge carriers and restrains the recombination of hole-electron pairs. This is significantly attributed to the graphene as a bridge of MoS2/TiO2 and the incorporation of graphene suggesting the synergistic effect of rapid electron-transferring of photoinduced electrons and holes, and the powerful electron-collecting of graphene suppressing charge recombination rate. The current work not only provides insight into graphene-bridged heterojunction photocatalytic for H2 evolution, but also motivates people to actively explore novel heterojunctions with various photocatalysts towards various applications. ”
- Give a descriptive schematic mechanism for the photocatalytic hydrogen evolution along with reaction step.
Reply: Thanks for the kind suggestion. It has been described in Page 7-8, Lines 433~448: “ According to the above results, a possible photocatalytic H2 production mechanism can be proposed (Fig. 7). During the visible light activity, photoexcited electrons in both TiO2 and MoS2 are induced from VB to their CB respectively, with holes moved on their VB. In addition, donated through the vigorous interface conjugation between TiO2 and MoS2, the electrons transport from the CB of TiO2 to the VB of MoS2 and suppress the photoexcited holes on the reason of a Z-scheme charge carriers transport process. Additionally, owing to the low Fermi level of graphene, the formed Schottky junction between MoS2 and TiO2 could further accelerate the charge transfer and inhibit the electron-hole recombination rate. Consequently, these photoinduced charges can be transferred to the surface and conduct the H2 production more successfully. The interfacial graphene, which acts as an interfacial bridge, can remarkably accelerate the separation of photoexcited electrons and holes, resulting in higher charge transfer efficiency between TiO2 and MoS2. In this work, we indicated that graphene as a conductive “bridge” can be improved the valid charge carriers separation efficiency and an useful interfacial charge transfer path. This result provides a basis for the realization of the design of photocatalysts for the H2 production, and for enhancing the stability of heterojunction to photocorrosion. ”
- Give a separate section in the text to discuss the photocatalytic hydrogen evolution mechanism in detail with references. This topic will be probably at the top portion of conclusion section.
Reply: Thanks for the kind suggestion. It has been described in Page 13, Lines 505~516: “ Therefore, the H2 production efficiency of MTG nanohybrids was obviously enhanced to 824.6 μmol-g-1-h-1 with visible light irradiation, which was 13.5 folds and 7.6 folds greater than that of pristine TiO2 (61.4 μmol-g-1-h-1) and MoS2 (109.6 μmol-g-1-h-1), respectively. The matched energy band structure between graphene and MoS2/TiO2, which enhances the separation ability of photo-induced charge carriers and restrains the recombination of hole-electron pairs. This is significantly attributed to the graphene as a bridge of MoS2/TiO2 and the incorporation of graphene suggesting the synergistic effect of rapid electron-transferring of photoinduced electrons and holes, and the powerful electron-collecting of graphene suppressing charge recombination rate. The current work not only provides insight into graphene-bridged heterojunction photocatalytic for H2 evolution, but also motivates people to actively explore novel heterojunctions with various photocatalysts towards various applications. ”
- In figure 6c and d, write the names of materials instead of digits.
Reply: Thanks for the kind suggestion. We have modified Figure 6c-d and shown in the revised manuscript.
Fig. 6. (c) H2 production of MTG -10 in TEOA with various pH values in three repeated tests. (d) H2 production with various amounts of MTG -10.
- You do not show the light. You used visible light or Uv-light. If you used visible light, then which filter cut you used. I do not see any information related. Repeat the whole paper properly.
Reply: Thanks for the kind suggestion. It has been described in Page 3, Lines 123~125: “ Visible light (410 < λ < 790 nm) was performed as an irradiation source, which was achieved by using both a 790-nm cutoff filter (short-wave-pass) and a 410-nm cutoff filter (longwave-pass). ”
- Show the effect of different sacrificial agent with your results.
Reply: Thanks for the kind suggestion. It has been described in Page 3 and 12, Lines 117~119 and 417~428: “ The types and concentrations of the sacrificial agent are 10 vol% triethanolamine (TEOA) and the equimolar amount of organic alcohol, including methanol, and ethanol. ” and “ Fig. 6f shows when the sacrificial agent was methanol and ethanol, the hydrogen production activity significantly decreased, with methanol resulting in the highest hydrogen production activity of only 384.3 μmol g−1 h−1, indicating that these alcohols had inferior hole-capture abilities compared with TEOA. H2 evolution rate over MTG-10 photocatalyst in methanol, ethanol and TEOA aqueous solutions were 384.3, 327.6 and 824.6 μmol g−1 h−1, respectively.”
Fig. 6 (f) photocatalytic H2 production activity over different sacrificial agents.
- Provide the important analysis such as surface area of all materials probably in form of table, photoluminescence spectra (PL) and DRS of materials. These parameters are important for one photocatalyst.
Reply: Thanks for the kind suggestion. We have added Table 1, modified Figures 6i and shown it in the revised manuscript. It has been described in Page 9-10, Lines 323~330: “ As a result, the massive Brunauer-Emmett-Teller (BET) surface areas of MoS2 (38.4 m2 g− 1 ), TiO2 (22.8 m2 g−1 ) and MTG-10 (70.2 m2 g−1 ) nanohybrids were evaluated, respectively (Table 1). The primary pore diameter of MoS2, TiO2, and MTG-10 were ~ 39.5, 49.7 and 9.2 nm, respectively. Furthermore, the pore volumes of MoS2, TiO2, and MTG-10 were ~0.195 cm3 g-1, 0.126 cm3 g-1and 0.386 cm3 g-1, respectively. BET of the MTG-10 sample was higher than 2~3 folds than that of MoS2 and TiO2, implying the introduced graphene had a significant effect on increasing the specific surface area of MTG photocatalysts. ”
Table 1. Summary of BET parameters of MoS2, TiO4, and MTG-10 photocatalysts.
|
Samples |
Specific surface area (m2/g, BET)a |
Total pore volume (cm3/g, BET)b |
Average pore diameter (nm, BJH)b |
|
MoS2 |
38.4±5 |
0.195±0.03 |
39.5±5 |
|
TiO2 |
22.8±5 |
0.126±0.03 |
49.7±5 |
|
MTG-10 |
70.2±5 |
0.386±0.03 |
9.2±2 |
a Received from BET analysis.
b Relative pressure (P/P0) was 0.99.
- Provide the FTIR analysis.
Reply: Thanks for the kind suggestion. We have included our partial XPS studies in Supporting Information and described some related information. It has been described in Page 6, Lines 232~237: “ The Fourier Transform Infrared (FT-IR) curves of MoS2, TiO2 and MTG-10 samples are characterized to check the interaction in heterostructures in Fig. 2l. The total patterns of samples are similar suggesting that the presence of graphene formation in MTG-10 samples does not mighty affect the basis construction of the MoS2 and TiO2. The feature peak of aromatic ring plane vibration could be noted from 1150 cm-1 to 1800 cm-1. ”
Fig. 2. (l) FT-IR of samples.
- Why you used 10 ml methanol with 100 ml of water. What is the role of methanol even you used the sacrificial agent TEOA in photocatalytic water splitting. Explain with facts.
Reply: Thanks for the insightful comment. It has been described in Page 3 and 12, Lines 117~119 and 417~428: “ The types and concentrations of the sacrificial agent are 10 vol% triethanolamine (TEOA) and the equimolar amount of organic alcohol, including methanol, and ethanol. ” and “ Fig. 6f shows when the sacrificial agent was methanol and ethanol, the hydrogen production activity significantly decreased, with methanol resulting in the highest hydrogen production activity of only 384.3 μmol g−1 h−1, indicating that these alcohols had inferior hole-capture abilities compared with TEOA. H2 evolution rate over MTG-10 photocatalyst in methanol, ethanol and TEOA aqueous solutions were 384.3, 327.6 and 824.6 μmol g−1 h−1, respectively.”
Fig. 6 (f) photocatalytic H2 production activity over different sacrificial agents.
- You detect the photocatalytic hydrogen evolution without the induction of co-catalyst?? In case show the effect of co-catalyst as comparison. In my opinion, such performance is not possible without co-catalysts specially Pt. All the materials you used have large bandgap and almost unactive under visible light. It is better that you provide the following parameters
(a) Band gap of materials
Reply: Thanks for the insightful comment. We have been revised and added in Page 5, Line 177-185: “ The band structures of the samples were estimated by the Kubelka-Munk (M-K) equation αhv = A(hv-Eg)1/2, and acquired the corresponding band gap energy (Eg) as exhibited in Fig. 1c, in which α, hv, A and Eg were the absorption coefficient, photon energy, a constant and direct band gap (eV), respectively. Besides, the band gap values of the as-fabricated heterojunction that were calculated by the UV–vis absorption edges via the M-K equation exhibited that 2.41 eV of MTG-2 was decreased to 2.26 eV of MTG-20 after increasing with graphene. Therefore, it is indicated that MTG nanohybrid photocatalysts possess favorable band gap energy to absorb solar light and approach the transition from H+ to H2. ”
Fig. 1 (c) Band gap plot of as-synthesized MTG nanohybrid photocatalysts with different amounts of graphene.
(b) Effect of co-catalyst in comparison with no co-catalyst performance.
Reply: Thanks for the kind suggestion. It’s true that our original statement is not clearly enough. According to your suggestion, we have added some additional description and cited some references in the revised manuscript. It has been described in Page 12~13, Lines 438~445: “H2 production performance of MTG-10 with 3.0 wt.% and 1.0 wt.% Pt as cocatalyst are 382.3 and 484.5 μmol g−1 h−1 in Fig. 6(g). When MTG-10 with 1.0 wt.% Pt, H2 production performance over MTG-10 with 3.0 wt.% Pt as cocatalyst. In addition, H2 production performance of MTG-10 without Pt of 824.6 μmol g−1 h−1 was approached. It is implied that MTG-10 indicates the optimal efficiency, and the suggested H2 production performance of MTG-10 is better than 2.2 and 1.7 folds of MTG-10 with 3.0 wt.% Pt and MTG-10 with 1.0 wt.% Pt, respectively. ”
Fig. 6(g) H2 evolution efficiency of MTG-10 with and without Pt.
- Double check the English corrections, grammar checks, pronunciation, punctuation, and other essential changes in the revised manuscript copy.
Reply: Thanks for the kind suggestion. The revised manuscript is checked by professional experts. We have carefully rechecked the entire manuscript and passed through the 'Grammarly’ online software to improve the revised manuscript's English language. We believe that the language of the revised manuscript can meet the journal's requirements.
- Some papers in the introduction portion need to be cited for good interest
https://doi.org/10.1016/j.mattod.2023.02.025
https://doi.org/10.1016/j.ensm.2023.102780
Reply: Thanks for the kind suggestion. The reference has been added, and changes have been incorporated into the revised manuscript.
Ref3 Hayat, A.; Sohail, M.; Jery, A. E.; Al-Zaydi, K. M.; Raza, S.; Ali, H.; Al-Hadeethi, Y.; Taha, T.A.; Din, I. U.; Khan, M. A.; Amin, M. A.; Ghasali, E.; Orooji, Y.; Ajmal, Z.; Ansari, M. Z. Recent advances in ground-breaking conjugated microporous polymers-based materials, their synthesis, modification and potential applications, Mater. Today 2023, 64, 180.
Ref 14 Hayat, A.; Sohail, M.; Jery, A. E.; Al-Zaydi, K. M.; Raza, S.; Ali, H.; Ajmal, Z.; Zada, A.; Taha, T.A.;I Din, . U.; Khan, M. A.; Amin, M A.; Al-Hadeethi, Y.; Barasheed, A. Z.; Orooji, Y.; Khan, J.; Ansari, M. Z. Recent advances, properties, fabrication and opportunities in two-dimensional materials for their potential sustainable applications, Energy Storage Mater. 2023, 59, 102780.

Reviewer 3 Report
The manuscript "A novel MoS2/TiO2/graphene nanohybrid for enhanced photo-2 catalytic hydrogen evolution under visible light irradiation " had been reviewed.
1. Introduction: there is no current information on MoS2/TiO2/graphene or MoS2/TiO2 photoactivity. What's new in proposed research?
2. Description of the synthesis is not clear to me.
3. Photoluminescence spectra, UV- vis absorbance and Electrochemical impedance spectroscopy for pristine MoS2, TiO2 and graphene should be investigated.
4. how the authors converted the peak areas into umol/g?
5. “The photocatalytic H2 evolution effect of MoS2, TiO2, and MTG nanohybrids was per-277 formed with visible light with triethanolamine (TEOA) as the sacrificial reagent.”
“The photocatalytic H2 generated activity of the as obtained nanohybrid was further measured by GC 7890 gas chromatography with a TCD detector from an aqueous solution including methanol as a sacrificial reagent.”
So which sacrificial reagent was used? TEOA or methanol?
6. “As ex-278 hibited in Fig. 6(a), the H2 production efficiency of MTG -10 improves by 7.6 and 13.5 times 279 those of MoS2 and TiO2 samples, respectively.” “When the 282 adding of graphene was excessive, the graphene aggregated on the surface of the sphere-283 like MTG nanohybrids, suggesting an increase in the surface rugged of MTG nanohybrids 284 and hence prohibiting the internal diffusion of the sacrificial reagent „Besides, exces-285 sive graphene may develop recombination centers for photoinduced charge carriers, then 286 decrease the photocatalytic activity. “ The activity for MTG -20 was the highest.
7. In my opinion XRD after cycle stability was different than before.
8. The correlation of the properties of photocatalysts with their photocatalytic activity should be expanded in the discission.
9. Mechanism of the photocatalyst excitation and hydrogen generation should be proposed in the discussion.
Author Response
Reviewer3
Comments and Suggestions for Authors
The manuscript "A novel MoS2/TiO2/graphene nanohybrid for enhanced photo-catalytic hydrogen evolution under visible light irradiation " had been reviewed.
- Introduction: there is no current information on MoS2/TiO2/grapheneor MoS2/TiO2 What's new in proposed research?
Reply: Thanks for the kind suggestion. It has been described in Introduction (lines 70-73): “ Recently, several works about the suitable and useful photo-assisted deposition process to well build MoS2/TiO2 heterojunction [20] and illustrated the great photocatalytic efficiency of MB and 2-CP removal with visible light have been published [21].”
[20] Teng, W.; Wang, Y.M.; Lin, Q.; Zhu, H.; Tang, Y.B.; Li, X.Y. Synthesis of MoS2/TiO2 nanophotocatalyst and its enhanced visible light driven photocatalytic performance, J. Nanosci. Nanotechnol. 2019, 19, 3519.
[21] Zhu, Y.Y.; Ling, Q.; Liu, Y.F.; Wang, H.; Zhu, Y.F. Photocatalytic H2 evolution on MoS2-TiO2 catalysts synthesized mechanochemistry, Phys. Chem. Chem. Phys. 2014, 2, 933.
- Description of the synthesis is not clear to me.
Reply: Thanks for the kind suggestion. We have done support the statements and more specific on the Experimental Section in Page 2, Line 92-93 and Page 13, Line 1-3: “ The fabrication of nanohybrids and hydrothermal process was illustrated in Scheme 1. ” . It has been described in Page 3, Lines 103~105: “ Next, the mixed suspension was transferred to a 50 mL Teflon-lined stainless-steel autoclave for hydrothermal treatment at 180 â—¦C for 16 h. ”
Scheme 1. The schematic illustration of the synthesis process of photocatalysts.
- Photoluminescence spectra, UV- vis absorbance and Electrochemical impedance spectroscopy for pristine MoS2, TiO2 and graphene should be investigated.
Reply: Thanks for the kind suggestion. We have modified Figures and shown in the revised manuscript.
Fig. 4 (a) Photoluminescence (PL) spectra. Fig. 1b UV- vis absorbance of the samples.
Fig. 4 (b) Electrochemical impedance spectroscopy (EIS).
- how the authors converted the peak areas into umol/g?
Reply: Thanks for the kind suggestion. The photocatalytic activity of photocatalysts was evaluated by measuring the hydrogen evolution amount from the organic aqueous (methanol, ethanol and TEOA) solution by an on-lined gas chromatography (GC). The hydrogen concentration was measured by a GC chromate graph equipped with a TCD (120 °C, 60 mA) and a Molecular Sieve (120 °C) column.
Fig. 6. H2 production of (a) MoS2, TiO2, and MTG nanohybrids (MTG -2, MTG -5, MTG -10, and MTG -20).
- “The photocatalytic H2 evolution effect of MoS2, TiO2, and MTG nanohybrids was performed with visible light with triethanolamine (TEOA) as the sacrificial reagent.”
“The photocatalytic H2 generated activity of the as obtained nanohybrid was further measured by GC 7890 gas chromatography with a TCD detector from an aqueous solution including methanol as a sacrificial reagent.”
So which sacrificial reagent was used? TEOA or methanol?
Reply: Thanks for the insightful comment. It has been described in Page 3 and 12, Lines 117~119 and 417~428: “ The types and concentrations of the sacrificial agent are 10 vol% triethanolamine (TEOA) and the equimolar amount of organic alcohol, including methanol, and ethanol. ” and “ Fig. 6f shows when the sacrificial agent was methanol and ethanol, the hydrogen production activity significantly decreased, with methanol resulting in the highest hydrogen production activity of only 384.3 μmol g−1 h−1, indicating that these alcohols had inferior hole-capture abilities compared with TEOA. H2 evolution rate over MTG-10 photocatalyst in methanol, ethanol and TEOA aqueous solutions were 384.3, 327.6 and 824.6 μmol g−1 h−1, respectively.”
Fig. 6 (f) photocatalytic H2 production activity over different sacrificial agents.
“As exhibited in Fig. 6(a), the H2 production efficiency of MTG -10 improves by 7.6 and 13.5 times those of MoS2 and TiO2 samples, respectively.” “When the adding of graphene was excessive, the graphene aggregated on the surface of the sphere- like MTG nanohybrids, suggesting an increase in the surface rugged of MTG nanohybrids and hence prohibiting the internal diffusion of the sacrificial reagent „Besides, excessive graphene may develop recombination centers for photoinduced charge carriers, then decrease the photocatalytic activity. “ The activity for MTG -20 was the highest.
Reply: Thanks for the kind suggestion. As displayed in Fig. 6 (a), the degradation efficiency of MTG nanohybrid was significantly improved than that of MoS2 and TiO2 under visible light irradiation. With the increase of graphene content, the photocatalytic performance of ternary heterojunction enhanced to a maximum value, after which it decreased with the redundancy of graphene. It could be distinctly seen that the highest photocatalytic H2 production of approximately 824.6 μmol g−1 h−1 was achieved by MTG-20 samples. MTG-20 nanohybrid was not displayed noticeable degradation performance towards H2 production efficiency (only 0.02% enhancement) due to the limited light absorption [r6]. The phenomenon indicated that an appropriate amount of graphene coupling with TiO2 and MoS2 was conducive to improving the photocatalytic activity of photocatalysts. In this case, owing to its attractive conductivity, graphene could serve as an electron bridge mediator to facilitate photogenerated charge carrier separation in the photocatalyst [r6]. However, excessive introduction of graphene might cover some active sites of the photocatalysts and inhibit the light absorption of TiO2 and MoS2, leading to the decrease of electron-hole photoexcitation, thereby reducing the photocatalytic H2 production performance of the photocatalysts [r7].
Fig. 6. H2 production of (a) MoS2, TiO2, and MTG nanohybrids (MTG -2, MTG -5, MTG -10, and MTG -20).
r6 Zhou, J.; Liu, C.; Xia, L.; Wang, L.; Qi, C.; Zhang, G.; Tan, Z.; Ren, B.; Yuan, B. Bridge-graphene connecting polymer composite with a distinctive segregated structure for simultaneously improving electromagnetic interference shielding and flame-retardant properties, Colloids Surf. A: Physicochem. Eng. Asp. 2023, 661, 130853.
r7 Zheng, S.; Li, Y.; Hao, J.; Fang, H.; Yuan, Y.; Tsai, H. S.; Sun, Q.; Wan, P.; Zhang, X.; Wang, Y. Hierarchical assembly of graphene-bridged SnO2-rGO/SnS2 heterostructure with interfacial charge transfer highway for high-performance NO2 detection, Applied Surface Science 568 (2021) 150926.
- In my opinion XRD after cycle stability was different than before.
Reply: Thanks for the kind suggestion. In Fig. 6h, the crystal structure of MTG-10 photocatalysts before and after use was measured by XRD analysis. The positions of XRD peaks (circular dotted line) stayed the same after five cycles. It could be apparently observed that there was no noticeable change of peak and phase for the recycled photocatalysts, which also suggested the high stability of MTG nanohybrids. According to the above analysis, MTG nanohybrids possessed excellent stability and reusability, and had good promise for efficient utilization.
Fig. 6 (h) XRD analysis of MTG -10 after and before the H2 production test.
- The correlation of the properties of photocatalysts with their photocatalytic activity should be expanded in the discission.
Reply: Thanks for the kind suggestion. It has been described in Page 13, Lines 461~464: “ In this work, we indicated that graphene as a conductive “bridge” can be improved the valid charge carriers separation efficiency and an useful interfacial charge transfer path. This result provides a basis for the realization of the design of photocatalysts for the H2 production, and for enhancing the stability of heterojunction to photocorrosion. ”
- Mechanism of the photocatalyst excitation and hydrogen generation should be proposed in the discussion.
Reply: Thanks for the kind suggestion. It has been described in Page 13, Lines 449~464: “ According to the above results, a possible photocatalytic H2 production mechanism can be proposed (Fig. 7). During the visible light activity, photoexcited electrons in both TiO2 and MoS2 are induced from VB to their CB respectively, with holes moved on their VB. In addition, donated through the vigorous interface conjugation between TiO2 and MoS2, the electrons transport from the CB of TiO2 to the VB of MoS2 and suppress the photoexcited holes on the reason of a Z-scheme charge carriers transport process. Additionally, owing to the low Fermi level of graphene, the formed Schottky junction between MoS2 and TiO2 could further accelerate the charge transfer and inhibit the electron-hole recombination rate. Consequently, these photoinduced charges can be transferred to the surface and conduct the H2 production more successfully. The interfacial graphene, which acts as an interfacial bridge, can remarkably accelerate the separation of photoexcited electrons and holes, resulting in higher charge transfer efficiency between TiO2 and MoS2. ”
Fig. 7. Illustration of charge transfer and H2 precipitation process in MTG nanohybrids under light irradiation.
